# Inequality of opportunity in selection procedures limits diversity in higher education: An intersectional study of Dutch selective higher education programs

**Lianne Mulder**[1,2]*, **Eddymurphy U. Akwiwu**[3], **Jos W. R. Twisk**[3], **Andries S. Koster**[4], **Jan Hindrik Ravesloot**[5], **Gerda Croiset**[6], **Rashmi A. Kusurkar**[1,2,7], **Anouk Wouters**[1,2]

**1** Amsterdam UMC Location Vrije Universiteit Amsterdam, Research in Education, Amsterdam, The Netherlands, **2** LEARN! Research Institute for Learning and Education, Faculty of Psychology and Education, Vrije Universiteit Amsterdam, Amsterdam, The Netherlands, **3** Amsterdam UMC, Vrije Universiteit Amsterdam, Epidemiology and Data Science, Amsterdam Public Health, Amsterdam, The Netherlands, **4** Department of Pharmaceutical Sciences, Utrecht University, Utrecht, The Netherlands, **5** Department of Medical Biology, Amsterdam UMC, University of Amsterdam, Amsterdam, Netherlands, **6** Wenckebach Institute for Education and Training, University Medical Center Groningen, Groningen, The Netherlands, **7** Amsterdam Public Health, Quality of Care, Amsterdam, The Netherlands

* l.m.a.mulder@amsterdamumc.nl

**Data Availability Statement:** The OSF link (https://osf.io/rwhg3/?view_only=

## Abstract

Selection for higher education (HE) programs may hinder equal opportunities for applicants and thereby reduce student diversity and representativeness. However, variables which could play a role in inequality of opportunity are often studied separately from each other. Therefore, this retrospective cohort study conducts an innovative *intersectional* analysis of the inequality of opportunity in admissions to selective HE programs. Using a combination of multivariable logistic regression analyses and descriptive statistics, we aimed to investigate 1) the representativeness of student populations of selective HE programs, as compared to both the applicant pool and the demographics of the age cohort; 2) the demographic background variables which are associated with an applicant's odds of admission; and 3) the intersectional acceptance rates of applicants with *all*, *some* or *none* of the background characteristics positively associated with odds of admission. The study focused on all selective HE programs (n = 96) in The Netherlands in 2019 and 2020, using Studielink applicant data (N = 85,839) and Statistics Netherlands microdata of ten background characteristics. The results show that student diversity in selective HE programs is limited, partly due to the widespread inequality of opportunity in the selection procedures, and partly due to self-selection. Out of all ten variables, migration background was most often (negatively) associated with the odds of receiving an offer of admission. The intersectional analyses provide detailed insight into how (dis)advantage has different effects for different groups. We therefore recommend the implementation of equitable admissions procedures which take intersectionality into account.

7114b77c9f8b4062ab31f885ce331a65) is now public will remain public forever. The rest of the data availability statement is accurate.

**Funding:** This work was supported by a Microdata Access Discount award from ODISSEI, the Open Data Infrastructure for Social Science and Economic Innovations (https://ror.org/03m8v6t10) (awarded to LM). The funders had no role in study design, data collection and analysis, decision to publish, or preparation of the manuscript.

**Competing interests:** The authors have declared that no competing interests exist.

## Introduction

Globally, the odds of entering higher education (HE) are neither divided equally nor equitably amongst all adolescents [1, 2]. There are several sociodemographic characteristics which, around the world, can result in barriers to accessing HE. These relate to socio-economic status (SES), ethnic/migration background, parental education and occupation, age, sex and having an urban versus rural background. However, these factors are often studied separately from each other, instead of combining elements of applicants' identities in one analysis (e.g., analyzing outcomes by combining sex, ethnic background, and age). Therefore, this retrospective cohort study conducts an innovative *intersectional* analysis of the inequality of opportunity in admissions to selective HE programs. By doing so, we aim to improve intersectional insights into educational (dis)advantage in selective admissions, which affects the diversity of the matriculating student population. As a diverse and representative student population is essential to achieving a better quality of education [3–5], stimulating improved representation is crucial. Furthermore, a diverse classroom prepares students for their role as professional in a labor market where they will work with patients, customers, clients, and colleagues from a wide range of backgrounds [6]. Additionally, for certain labor sectors, such as healthcare, it is essential to educate a diverse workforce to be able to provide the best care to a diverse patient population [7].

Intersectionality is a theory developed by Kimberlé Crenshaw [8] which holds that identities are multi-layered. On each layer, a person can occupy a position which is privileged in their particular society and seen as 'the Norm', or a position which is disadvantaged, and seen as the non-normative 'Other'. This places an individual on multiple hierarchical axes of privilege/oppression relating to social structures [8–10]. Each axis, therefore, may relate to structural advantages or disadvantages of students in the educational sphere. From an intersectional point of view, a person's experience cannot be understood sufficiently by considering these layers or axes independently [11]. Instead, the combination of their position on these axes influences a person's experience and opportunities in a given context. Using intersectionality theory to analyze the inequality of opportunity in HE selection is important, because applicants to selective HE programs can belong to multiple groups which are disadvantaged in the education system. For example, they may have a lower SES *and* an underrepresented migration background *and* be 'social newcomers' [12] in their chosen field of study (meaning: no social ties such as parents in that field). These applicants could thereby face multiple barriers in the route to entering a selective HE program, such as in the case of health professions education [13]. Similarly, applicants without a migration background, from high SES families, who are 'insiders' in the field (e.g. they have parents who practice the same profession as the program is intended to train the student in), may experience accumulated advantages which are not earned on the basis of their own merit [14]. However, the size and significance of these combined (dis)advantages are unknown. This study aims to fill that gap in the literature.

### Background characteristics and educational inequality

On the level of individual background characteristics, there are several general patterns of educational inequities. For example, students from socio-economically disadvantaged backgrounds are known to face numerous obstacles on the route to HE [15], including disadvantages in competitive selection procedures [3, 16, 17]. Applicants from a low SES background are often already underrepresented in the applicant pools of competitive HE programs [18]. There, they find themselves contending against applicants from high SES backgrounds, who had more resources to adequately prepare them [1]. Oftentimes low SES applicants lose this competition,

not because they are less qualified, but because the selection procedures are (unintentionally) biased against them [19–21]. This is a major problem, as socio-economic conditions are likely the most important underlying factor influencing educational inequalities [1, 22].

Parental occupations may also influence odds of HE participation [1] or odds of admission in a selection process. For example, students with at least one parent in a Science, Technology, Engineering, and Mathematics (STEM) related job show higher levels of STEM interest, motivation and achievement, and choice to major in a STEM field [23]. Their higher achievement levels [23] are likely to result in higher odds of admission in selection procedures which give weight to previous scholastic achievement. Another example are the higher odds of admission to health professions education programs for applicants with at least one parent who is a registered healthcare professional [13]. Interviews with high school students who were interested in health professions education programs also showed how a social network connection in the medical field helped students to develop their interest for such programs, and why a network connection was the most important facilitator in preparing for the selection procedure [24].

Furthermore, having an underrepresented ethnic or migration background is associated with educational disadvantage and barriers to optimal achievement due to e.g. microaggressions and racism [25], lower expectations by the education system [26], the effects of testing language [27], and differential access to high-quality and highly-resourced schools [3, 26, 28]. As a consequence, in many countries, applicants with such backgrounds have lower odds of admission than their white peers without a migration background [13, 29–31].

Finally, rural applicants may be at a disadvantage in entering HE. For example, Canadian research has reported lower performance in a multiple mini interview of applicants graduating from a rural high school [32]. In China, educational stratification is mainly driven by a rural-urban divide in which rural schools are disadvantaged in numerous ways. For example, they receive lower levels of funding and have poorer teacher qualifications, negatively affecting school enrollment and educational attainment [33]. Elite universities have witnessed an increasing rural-urban enrollment disparity: In the 1970s, one in two first-year students were from a rural background; whereas in 2011, this was one in seven [34].

Altogether, the aforementioned examples of unequal access to HE represent a major problem for educational quality, as a diverse, representative student population has been shown to be associated with better educational experiences for students [3–5]. Furthermore, disadvantaging applicants who are already underrepresented in HE reproduces societal and educational inequities [14, 21]. Admissions committees should use valid selection methods [35], but research by the Dutch Inspectorate of Education suggests that this is not always the case [36]. However, it is unknown what the effects are of belonging to multiple (dis)advantaged groups at the same time. This study aims to address the gap in the literature of the intersectional (dis) advantages which applicants face in accessing selective programs in HE, based on nationwide data from The Netherlands.

## The context: Higher education in The Netherlands

In The Netherlands, the typical route to HE consists of several steps. In the transition from elementary to high school, around ages 11–12, pupils are placed in either the pre-vocational (vmbo, 4 years), higher general (havo, 5 years), or pre-university (vwo, 6 years) track. Research has shown that there is inequality of opportunity in the placement into these tracks, and that the capabilities and potential of certain groups of children (e.g. lower SES, with certain migration backgrounds, from rural areas) are underestimated [37–39]. Although transitioning between tracks is possible, this is an exception [26], and the rates at which this happens differ greatly between provinces [40]. This means that for many children, their educational route is

determined at an age which is lower than in many other countries [39]. This contributes to inequality of opportunity [39].

In the final year of high school, pupils can apply to post-secondary education programs. Most higher general pupils enter HBO, but they could also enter vocational education. Most pre-university pupils enter university, but they too have the option to enter HBO or vocational education. Of these post-secondary education programs, only higher education institutions (HEIs) use selection to admit students. These include higher professional education (in Dutch: HBO–often referred to as university of applied sciences), and university. For sake of readability, we have used the Dutch abbreviation HBO in this manuscript.

Within HEIs, most study programs are not selective, but programs may use specific eligibility criteria. Applicants with other (non-traditional) educational backgrounds than specified in the eligibility criteria must prove that they meet equivalent educational levels [41]. For example, for many HBO programs, it is possible to apply with either a higher general diploma or a vocational education diploma. For many university programs, it is possible to apply with either a pre-university diploma, or with a completion certificate of all first-year courses of a (related) HBO study program called *propedeuse*.

Some study programs have to use selection procedures, due to capacity limitations called *numerus fixus* (fixed number). When there are more eligible applicants than seats, institutions must use at least two types of qualitative selection criteria (e.g. intelligence, motivation, study skills) to determine who will be offered a place in the program [42]. In the selection procedure, diversity may be lost, which could result in underrepresentation of certain groups. However, apart from four university health professions programs for which inequalities based on sex, migration background, socio-economic status and having healthcare professionals as parents were found [13], it is unknown to what extent this is the case across the entire HE domain. Therefore, we wanted to investigate the possible inequality of opportunity in all selective HE programs. We hypothesized that there could be study programs with less or no inequality of opportunity, and programs in which underrepresented applicants have higher odds of admission. Programs with proven inequality of opportunity in the selection procedure, could then learn from the programs who offer equal or equitable opportunities to applicants. Additionally, based on the importance of a representative workforce in certain labor markets (e.g. healthcare), we were interested in whether there are study programs where potential best practices could be found in admitting underrepresented applicant groups. Thus, we aimed to answer the following research questions:

1. How representative are student populations of selective HE programs, compared to: a) the applicant pool; b) the higher general population and pre-university population; and c) their age cohort?

2. Which demographic background variables (e.g. sex, migration background, year of birth, socio-economic status, parental work sector) are associated with an applicant's odds of admission into selective HE programs?

3. What are, from an intersectional perspective, the acceptance rates of applicants with *all*, *some* or *none* of the background characteristics positively associated with odds of admission?

## Methods

For this study, we used the STROBE cohort reporting guidelines [43] and the SAMPL guidelines [44]. These are statistical and methodological guidelines suited to the type of study we executed.

### Study design, setting and eligibility criteria, study size

We conducted a retrospective multi-cohort study using anonymized non-public microdata from Statistics Netherlands, focusing on applicants and students in academic years 2019–2020 and 2020–2021. These years were chosen, as they were the two most recent years for which essential variables were available in Statistics Netherlands microdata at the start of the research. Statistics Netherlands does not allow for analysis on single-institution data. Therefore, we created several clusters of studies which each consisted of at least three institutions, to make data non-traceable to any particular institution. In cases where different studies had to be combined to achieve this, clusters were made on the basis of similarity of programs (e.g., related to personal healthcare or not). Table 1 contains information on the study programs in all HBO clusters (starting with 'SH') and all university clusters (starting with 'SU').

The six HBO clusters included 19 HBO study programs, the nine university clusters included 28 study programs. We only included study programs taught at HEIs which are publicly funded and which are members of either the Netherlands Association of Universities of Applied Sciences (*Vereniging Hogescholen*) or Universities of the Netherlands (*Universiteiten van Nederland*), as these are the organizations which granted us permission to perform data analysis on their anonymized applicant pool and student population. It is important to note that many selective programs (e.g. Nursing) are only selective at one or a few HEIs, but that students can enroll in these programs at other HEIs without having to go through a selection procedure. Therefore, S1 Table also contains information on the number of HEIs that did not use selection for the particular study programs. Since this research focuses on inequality of opportunity in selection, the student populations of non-selective programs are not included in our analyses. We only used the term 'selective program' for programs which have a capacity limit and use selection to decide which applicants to admit for the limited places available. We created the following groups:

**Applicants for selective programs (N = 85,839).** This group consists of all domestic applicants who applied through Studielink (the national online application portal for post-secondary education in The Netherlands) for at least one selective program, and who received a ranking number. The group is divided into HBO applicants and university applicants. Applicants who never received a ranking number (e.g., because they decided to withdraw from the selection procedure, or were not eligible) were not included, since the selection committee was not able to offer them a place (see Fig 1).

**Age cohort (N = 204,075).** To compare the demographics of study clusters with the demographic characteristics of the young population, we used Statistics Netherlands microdata of all 16-year-olds who were registered in The Netherlands on October 1st in 2015. Within the age cohort, we also distinguished whether 16-year-olds were higher general or pre-university students, since these are the most common educational routes to HBO and university.

### Data sources

Studielink provided all applicant data of all selective HE programs from 2019 and 2020 to Statistics Netherlands, which pseudonymized the Citizen Service Numbers (CSN) of domestic applicants. The researchers used pseudonymized CSN to merge other Statistics Netherlands microdata to the dataset. International applicants were excluded from this research.

**Table 1. Information on clusters of study programs.**

| Cluster | HBO program | HBOs using selection | Average acceptance rates 2019 | Average acceptance rate 2020 |
|---|---|---|---|---|
| SH1 | Allied Medical Care | 3 | 52.9% (1028/1942) | 52.2% (994/1904) |
| | Nursing | 1 | | |
| | Midwifery | 3 | | |
| SH2 | Dental Hygiene | 4 | 36.5% (539/1477) | 38.6% (529/1372) |
| | Denturism | 1 (2020) | | |
| | Optometry | 1 | | |
| SH3 | Biology and Medical Laboratory Research | 2 (2019), 1 (2020) | 72.6% (864/1190) | 64.6% (674/1044) |
| | Forensic Science | 2 | | |
| | Medical Imaging and Radiation Therapy | 1 | | |
| SH4 | Physiotherapy | 6 | 78.5% (2263/2881) | 74.1% (2214/2987) |
| | Psychomotoric Therapy/Psychomotricity | 1 | | |
| | Sport Studies | 1 | | |
| SH5 | Creative Media and Game Technologies | 1 | 68.9% (1081/1570) | 70.0% (996/1422) |
| | Fashion & Textile Technologies | 1 | | |
| | Industrial Design Engineering | 1 | | |
| | Art and Economics | 1 (2020) | | |
| SH6 | Applied Psychology | 4 | 63.8% (2041/3199) | 54.5% (1719/3157) |
| | Applied Biology | 1 | | |
| | Skin Therapy | 2 | | |
| Cluster | University program | Universities using selection | | |
| SU1 | Medicine | 8 | 45.4% (2986/6571) | 43.0% (2969/6912) |
| SU2 | Dentistry | 4 | 41.6% (717/1725) | 38.1% (643/1689) |
| | Pharmacy | 1 | | |
| SU3 | Psychobiology | 1 | 85.1% (4631/5439) | 75.8% (6845/9030) |
| | Psychology | 6 (2019), 9 (2020) | | |
| SU4 | Biomedical Sciences | 4 | 57.4% (1391/2425) | 48.1% (1068/2219) |
| | Biomedical Engineering | 1 (2019) | | |
| | Clinical Technology | 2 | | |
| SU5 | Biology | 1 | 59.3% (902/1521) | 46.9% (555/1184) |
| | Biotechnology | 1 | | |
| | Nutrition and Health | 1 | | |
| | Veterinary Medicine | 1 | | |
| | Nanobiology | 1 | | |
| SU6 | Artificial Intelligence | 4 | 81.1% (1421/1753) | 86.2% (1469/1704) |
| | Industrial Design | 2 | | |
| SU7 | Architecture, Urbanism & Building Sciences | 2 | 74.5% (2838/3810) | 68.9% (2411/3498) |
| | Mechanical Engineering | 1 | | |
| | Aerospace Engineering | 1 | | |
| | Computer Science & Engineering | 2 | | |
| | Global Sustainability Science | 1 | | |
| SU8 | Business Administration | 1 (2019) | 83.0% (3100/3737) | 78.5% (2673/3406) |
| | International Business | 2 | | |
| | International Business Administration | 1 | | |
| | Tax Law | 1 | | |
| | Industrial Engineering & Management Science | 1 | | |
| SU9 | International Relations and International Organization | 1 | 66.1% (1609/2436) | 60.0% (1602/2672) |
| | Political Science | 1 | | |
| | Criminology | 3 | | |

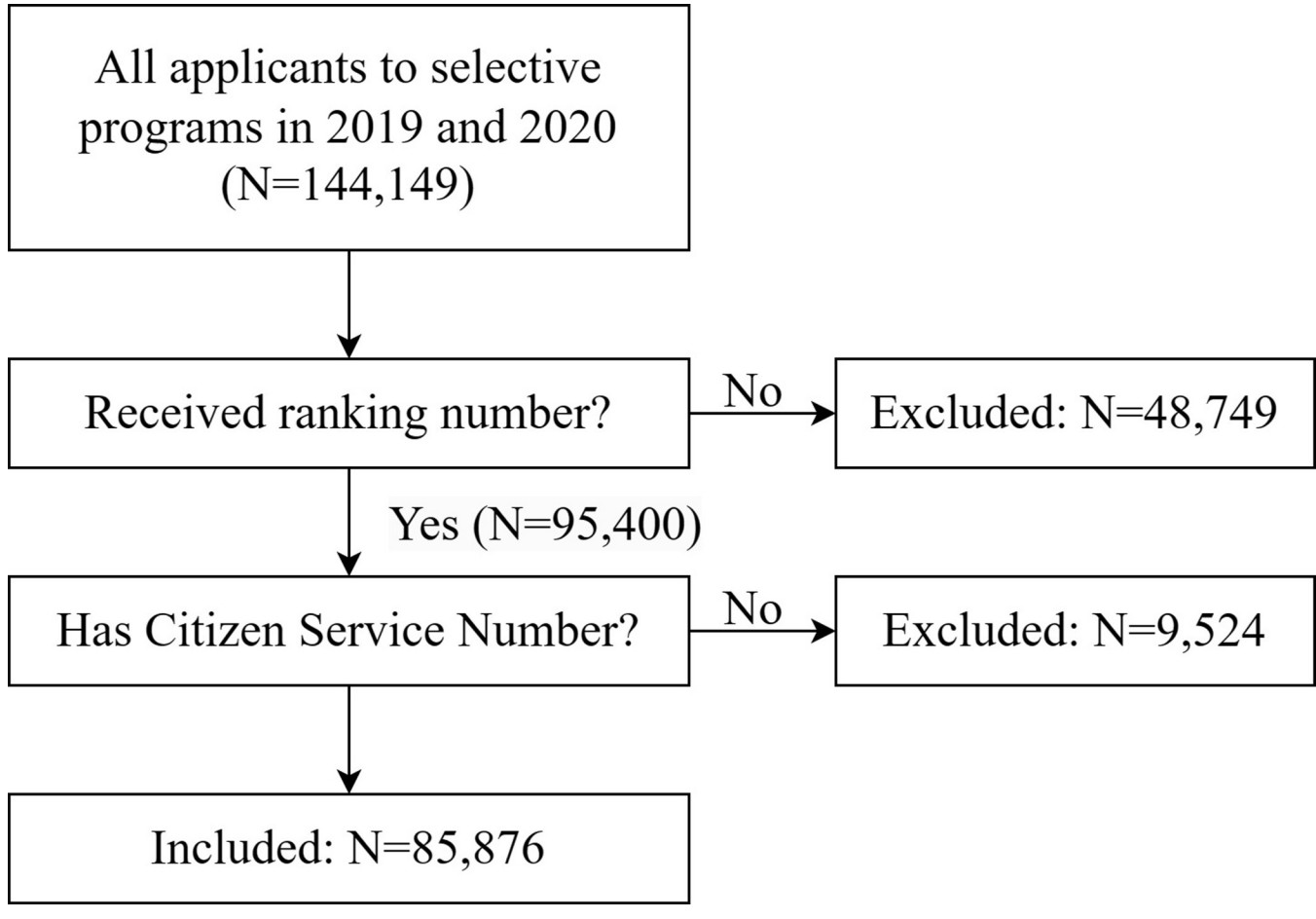

**Fig 1. Study size and exclusion criteria.**

**Ethics and replicability.**   The research project was based on anonymized data from Statistics Netherlands and Studielink. Therefore, participant consent was not required. Statistics Netherlands microdata and Studielink data are non-public. The statistical results comply to all Statistics Netherlands privacy regulations and the Dutch law regarding use of their non-public microdata (Wet op het Centraal Bureau voor de Statistiek, 2004). Selected data from Studielink of the applicant pool was anonymized by Statistics Netherlands. The researchers had no access to identifiable information.

This study could be replicated if access to the datasets is gained. Our research protocol, including all procedures, sources of variables, and software syntax for statistical analysis, can be found in S2 File and on https://osf.io/rwhg3/?
view_only = 7114b77c9f8b4062ab31f885ce331a65. Two government documents with data on HEIs and study programs were used to cross-check the accuracy of cluster sizes (number of institutions per cluster) in S1 Table. These can be found in the same database.

**Data of small groups.**   Statistics Netherlands regulations prescribe that descriptive statistics and analyses performed on groups smaller than 10 persons cannot be published in detail. As a consequence, we had to replace frequencies between 0–4 by '<5' and frequencies between 5–9 by '<10', to avoid traceability to an individual or small groups of individuals. If one

category within a variable had <5 or <10 persons, the next smallest category also had to be rounded to the nearest pentad (e.g. <15, <20).

**Variables.** The study includes the following applicant characteristics: sex, year of birth, migration background, country of birth, and degree of urbanity of postal code of address in 2015. It also includes the following applicants' parental data: Income percentile of parent with highest income; household assets percentile; number of parents who receive social welfare; number of parents who receive a social services income (excl. social welfare); number of parents who are registered healthcare professionals (HP) according to the official Dutch *BIG-register*; and the number of parents who work in the primary, secondary or vocational education sector. Details on these variables are described in Table 2. The Studielink dataset also contained information on whether an applicant had received an offer of admission, whether this offer was accepted, and whether an applicant had enrolled in the program by October 1st in the year of application.

**Bias.** We followed Šimundić's [54] classifications of bias in research to study potential sources of bias in advance of the analyses. By including *all* selective programs in HE, and an age cohort of *all* 16-year-olds on October 1st, 2015 as reference group for the demographics of the young population, we aimed to eliminate potential bias in sampling.

**Statistical analysis.** To determine the representativeness of each cluster's student population compared to the applicant pools, age cohort, and all first-year students in HBO and university, we used frequency tables to compare the distribution of students on all variables. Not all applicants who received a placement offer became students in that program. There can be several reasons for this. For example, they may have received an offer of admission in two different study clusters, and only accepted one. They could also decide to enter a non-selective study, or refrain from entering HE. It is also possible that after a student accepted an offer of placement, they failed their final exams in high school. This excludes them from entering HE.

After that, we examined data for evidence of multicollinearity amongst the independent variables using both the Pearson correlation coefficients between variables, and the variance inflation factor (VIF) of each variable. We then performed univariable logistic regression analyses to examine which of the independent variables were associated with being offered admission (the dependent variable). The results are shown with odds ratios (ORs) and the corresponding 95% confidence intervals (95% CIs). Statistical level of significance was set at 0.05. Multivariable logistic regression was used to create a model for each cluster in each academic year. This was done to investigate the effect of background variables of applicants on the odds of receiving an offer of placement, while adjusting for the other variables in the model. The results are shown with adjusted ORs and 95% CIs. We did not apply forward or backward selection. The regression analyses were performed on the applicants who received a ranking number.

Next, we used each cluster's multivariable model to create an intersectional analysis of the admission rates of two groups of applicants within that cluster: 1) applicants who possessed *all* demographic attributes which were significantly positively associated with the odds of being offered admission in the final model of that cluster (referred to as *group ALL*); and 2) applicants who possessed *none* of these demographic attributes (*group NONE*). If a variable had 3 categories (e.g. A, B, C) and only category B was shown to have significantly different odds of admission compared to reference group A, additional regression analyses were performed with reversed reference categories, to investigate if B and C also significantly differed from each other. The double analyses with different reference categories provided the information which of the three categories had significantly higher odds of admission.

For group ALL and NONE, we calculated the percentage of applicants which were offered admission. The comparison between these groups (and the ratio of the acceptance rate of

**Table 2. Demographic data recorded for each applicant and their parents.**

| Variable | Values | Rationale |
|---|---|---|
| **Appplicant data** | | |
| Sex | 0 = Male | The known male:female ratio differences in some fields of study [45] |
| | 1 = Female[a] | |
| Year of birth | Recoded in 3 categories to create groups of sufficient size for logistic regression analysis: | The known higher performance of older students (>21 years), compared to younger students, in Dutch medical education [46]. |
| | 0 = 1999 or earlier | |
| | 1 = 2000–2001 | |
| | 2 = 2002 or later | |
| Migration background | Statistics Netherlands original values are based on country of birth of the person, or the country of birth of parents. Each country has its own value, which we recoded in 5 categories for the descriptive statistics: | Earlier research showed the inequality of opportunity in selection to Dutch university health professions education programs for applicants with a Turkish, Moroccan, Surinamese and Dutch Caribbean migration background [13]. In this research, applicants with an Indonesian migration background constituted a group of similar size, with a similar (post-)colonial migration history as Surinamese and Dutch Caribbean migrants. |
| | 0 = No migration background | |
| | 1 = Turkish/Moroccan migration background | |
| | 2 = Surinamese/Dutch Caribbean/Indonesian migration background | |
| | 3 = EU/EEA/Swiss (European) migration background | |
| | 4 = Other migration background. | |
| | These were recoded in 3 categories for the regression analyses: | |
| | 0 = No migration background | |
| | 1 = Turkish/Moroccan/Surinamese/Dutch Caribbean/Indonesian (TMSDI) migration background | |
| | 2 = Other migration background (OMB) migration background | |
| Country of birth | 0 = Born in The Netherlands | |
| | 1 = Born abroad | |
| Degree of urbanity of postal code, of address in 2015 | Based on average number of addresses per $km^2$ of the area: | Urban areas may offer more opportunities to build one's CV and engage in extracurricular activities, which may be (an element of) a selection instrument used in The Netherlands [47]. A Canadian study showed lower performance in a multiple mini interview of applicants who graduated from a rural high school [32]. |
| | 1. Very strong (2500 or more) | |
| | 2. Strong (1500–2499); | |
| | 3. Average (1000–1499); | |
| | 4. Weak (500–999); | |
| | 5. Not at all (less than 500) | |
| | Recoded in 3 categories to create groups of sufficient size for logistic regression analysis: | |
| | 0 = (Very) strongly urban | |
| | 1 = Averagely urban | |
| | 2 = Weakly/not urban | |
| **Parental data** | | |
| Income percentile of parent with highest income | Scale of 1–100, recoded in 3 categories to create groups of sufficient size for logistic regression analysis: | The known barriers of low SES in the education field in general or in access to higher education, and the disproportionate share of students from high-income families among health professions education students [48–50]. Income and assets percentiles, rather than their values in euros, were included because percentiles indicate the relative position one occupies compared to the rest of the population. |
| | 0 = Percentile 1–70 | |
| | 1 = Percentile 71–90 | |
| | 2 = Percentile 91–100 | |
| Household assets percentile [b] | Scale of 1–100, recoded in 3 categories to create groups of sufficient size for logistic regression analysis: | |
| | 0 = Percentiles 1–40 | |
| | 1 = Percentiles 41–80 | |
| | 2 = Percentiles 81–100 | |

(*Continued*)

**Table 2.** (Continued)

| Variable | Values | Rationale |
|---|---|---|
| Number of parents who receive social welfare | 0, 1, 2, recoded in 2 categories to create groups of sufficient size for logistic regression analysis: | The low disposable income of families on social welfare, and the lack of a current network in the workforce, are hypothesized to have a negative effect on the child's opportunity to enter a selective program. |
| | 0 = 0 parents | |
| | 1 = 1 or 2 parents | |
| Number of parents who receive a social services income (excl. social welfare) | 0, 1, 2, recoded in 2 categories to create groups of sufficient size for logistic regression analysis: | There are different types of social services income, including types that are based on long-term illness or a labor disability. Combined with the lack of a current network in the workforce, these factors are hypothesized to have a negative effect on the child's opportunity to enter a selective program. |
| | 0 = 0 parents | |
| | 1 = 1 or 2 parents | |
| Number of parents who are registered healthcare professionals | 0, 1, 2, recoded in 2 categories to create groups of sufficient size for logistic regression analysis: | The known influence of having a network in the medical field as a facilitator in preparing for selection in health professions education programs [15, 16, 51–53] |
| | 0 = 0 parents | |
| | 1 = 1 or 2 parents | |
| Number of parents who work in the primary, secondary or vocational education sector | 0, 1, 2, recoded in 2 categories to create groups of sufficient size for logistic regression analysis: | We hypothesized that parents who work in the primary, secondary or vocational education sector can provide access to information about higher education and selection procedures. |
| | 0 = 0 parents | |
| | 1 = 1 or 2 parents | |

[a] It is acknowledged that not every individual is, or identifies as, 'male' or 'female', but Statistics Netherlands only has two possible sex categories. This means that e.g. intersex persons either have missing data on their sex registration, or are categorized as male or female. Transgender persons who have changed their sex registration in the national Personal Records Database are included in this study according to the sex which was registered in 2021.

[b] When parents live in different households, they may each have a different household assets percentile. In that case, we selected the highest percentile.

group ALL divided by that of group NONE) shows the contrast in opportunities for the groups that differed the most from each other. To provide insight into the acceptance rates of applicants who had *some*, but not all, of the characteristics positively associated with odds of admission, we created figures which showcase these rates for every combination of characteristics. All analyses were performed using IBM SPSS software for Windows, Version 25.0 (IBM Corp, Armonk, NY).

# Results

## Representativeness

S3 Table summarizes the demographic data of all applicants who received a ranking number, all applicants who were offered a seat in the program, and all those who became registered students, for each cluster. With these results, we can answer research question 1, on how representative student populations of selective HE programs are, compared to: a) the applicant pool; b) the higher general population and pre-university population; and c) their age cohort. Due to the extensive amount of descriptive statistics, we focus here on the variables with the largest differences between categories.

**Sex.** The results show that when student populations are not representative of their age cohort or the higher general/pre-university population, this is partially due to self-selection in applications. For example, in SH1 (healthcare related), men constituted 13–14.5% of the student population in 2019, respectively 2020. This is primarily due to them applying in lower numbers (approximately 16% of the applicant pool). Most clusters in which men are underrepresented, concern health professions education programs. SU7 (mostly consisting of engineering programs) was the only cluster in which women were significantly underrepresented: they made up 28–33% of the student population, but also only 28–31% of the applicant pool.

**Socio-economic status indicators.** When it comes to parental assets category, we see that compared to the age cohort, applicants and students from percentile category 1–40 (the 40% of households with the lowest assets levels) are underrepresented in *every* cluster except SH2. This is also the case when compared to the eligible pool (higher general for HBO programs, pre-university for university programs), except in SH2, SH6 and SU2.

The same can be observed for parental income category: compared to the age cohort, percentile category 1–70 is underrepresented in every cluster except SH2 and SU2. Compared to the eligible pool, students from this income category are underrepresented in every cluster except SH2, SH3_2019, SH6, SU1, and SU2. This means that in the majority of clusters, applicants and students who have parents belonging to the top-30% of the income distribution are disproportionately present. This is especially the case for cluster SU4 (Biomedical Sciences, Biomedical Engineering & Clinical Technology), where more than 85% of students have parents in the top-30% income category, and cluster SU1 (Medicine), where this is the case for at least 78% of students.

Compared to the age cohort, students with parents who receive social welfare are underrepresented in all clusters except SH2 and SU2. Relative to the eligible pool, they are underrepresented in all clusters except SH2, SH3, SH6, SU1, SU2, and SU3_2020.

**Migration background.** Compared to the age cohort, students without a migration background are overrepresented in SH1, SH4, SU4, and SU5. Students with certain migration backgrounds, such as Turkish or Moroccan, Surinamese, Dutch Caribbean or Indonesian, are underrepresented in the majority of clusters. This pattern is less evident when compared to the eligible pools. Students with an EU/EEA/Swiss migration background are overrepresented in most university clusters, both compared to the age cohort and the pre-university population. Finally, students with an Other migration background are sometimes (slightly) underrepresented, but more often overrepresented compared to the age cohort and eligible pools.

## Variables associated with odds of admission

Research question 2 aimed to find out which demographic background variables are associated with an applicant's odds of admission into selective HE programs. We excluded the variable 'born in The Netherlands or abroad' from our logistic analyses, as the variable 'migration background' is also based on (parental) country of birth. We used 'migration background' as it is able to give more detailed information on someone's background. However, the variable 'born in The Netherlands or abroad' remained in the descriptive statistics (S3 Table), to provide a better insight into those applicants who, according to the Statistics Netherlands definition, have a 'migration background' despite being born in The Netherlands. Their 'migration background', as defined by Statistics Netherlands, is then based on the country of birth of at least one of their parents. No evidence was found for multicollinearity in the remaining variables (the highest Pearson correlation was 0.389). S2 Table contains the univariable logistic regression results for each cluster, and S4 Table gives the multivariable regression models.

The *univariable* logistic regression results show that only in study cluster SU6_2020 (4 Artificial Intelligence programs; 2 Industrial Design programs), none of the individual variables were significantly associated with the odds of admission. All other study clusters showed significantly different odds based on at least one variable. The variables most often associated with the odds of admission, were migration background (23 out of 30 clusters), assets category (20 out of 30 clusters), and income category, sex and urbanity degree (all 17 out of 30 clusters). The variable associated the least often, was having 1 or 2 parents working in the primary, secondary or vocational education sector (6 out of 30 clusters).

After adjusting for all other variables, the picture of inequality of opportunity changes only slightly. The *multivariable* regression models in S4 Table show that the inequality of opportunity was widespread. SU6_2020 remained the only cluster that did not show significant differences in odds of admission between applicants. The variables which most often influenced the odds of receiving an offer of placement, were migration background, sex, and age. However, the groups within these variables which had higher odds varied between clusters. For example, in seven out of 15 clusters, male applicants had higher odds, whereas in eight out of 15 clusters, women had higher odds. These mixed results were also visible for the urbanity degree of the postal code where applicants lived in 2015, as in some cases, those from (very) strongly urban areas had higher odds, whereas in other clusters, this was the case for applicants from weakly/ non urban areas. In clusters where year of birth mattered, younger applicants (born in 2000 or later) were usually at an advantage.

Where groups are underrepresented based on sex, the inequality of opportunity in selection sometimes (slightly) compensates for their underrepresentation, whereas in others, it contributes to it further. In some clusters, however, there is a loss of diversity *after* offers of admission have been made as well. For example, in SH5_2019, women make up 55% of the applicant pool, 62.4% of the applicants who have been offered admission, but only 50.3% of the enrolled students. We do not have an explanation for this large difference.

In almost all clusters where migration background was significant, applicants without a migration background had higher odds (in 18 clusters). However, in one cluster (SH3_2020), applicants with a Turkish, Moroccan, Surinamese, Dutch Caribbean, or Indonesian (TMSDI) migration background had higher odds, and in one cluster (SU5_2020), applicants with an Other migration background had higher odds than applicants without a migration background. The multivariable models show that inequality based on migration background cannot be explained by the other nine variables included in the model: in the large majority of clusters, migration background remained significant after adjusting for those other background characteristics of applicants.

Analyses of the descriptive statistics shows that in 12 clusters where applicants without a migration background had higher odds, they were actually underrepresented in the applicant pool as compared to the higher general or pre-university population. Four of those clusters concerned health professions education programs. In some cases, the inequality of opportunity in admissions is the main reason for underrepresentation amongst students. For example, in SH1_2019 (Allied Medical Care, Nursing & Midwifery), 6% of applicants (117/1942) had a Turkish or Moroccan migration background, which is almost equal to their proportion in the age cohort (6.7%). However, these applicants only made up 1.3% of applicants offered admission (13/1028). Out of the 491 students who finally enrolled, less than 10 had a Turkish or Moroccan migration background. If enrolment rates had been equal to application rates, then there would have been 30 students in SH1_2019 with such a background.

In clusters where parental assets category was significant, those from higher categories consistently had higher odds. This was also the case for having healthcare professional parents, and parents who worked in the education sector. Having parents with either a social welfare or other social services income was negatively associated with odds of admission twice.

## Intersectional acceptance rates

To answer the third research question, we calculated the acceptance rates of applicants with either *all*, *some* or *none* of the background characteristics positively associated with odds of admission, based on the final regression models in S4 Table. Table 3 shows the results of this intersectional analysis for each study cluster. For example, the row of cluster SH1_2020 shows

**Table 3. Intersectional analysis of the inequality of opportunity in Dutch higher education selection procedures: Comparison between group ALL\* and group NONE\*\*.**

| Cluster | Sex | Age, based on Y. O.B. | Migration background | Urbanity degree of postal code in 2015 | Income category | Assets category | Parents on social welfare | Parents with SSI | HP parents | Parents working in PSVE sector | Acceptance rate of group ALL\* | Acceptance rate of group NONE \*\* | Ratio accept. rate ALL/ NONE | p-value of the ratio[a] |
|---|---|---|---|---|---|---|---|---|---|---|---|---|---|---|
| | | | Background characteristics associated with significantly higher odds of admission in each multivariable logistic regression model | | | | | | | | | | | |
| Total dataset | M | 2000 or later | None | | 71–100 | 41–100 | None | | | 1 or 2 | 69.7% (975/1398) | 34.9% (290/832) | 2,00 | < 0.001 |
| SH1_2019 | F | 1999 or earlier | None | AWNU | 1–90 | | | | 1 or 2 | 1 or 2 | >97% (data hidden) [b] | 11.1% (data hidden) [b] | 8.74–9.01 [c] | [c] |
| SH1_2020 | F | 2001 or earlier | None | AWNU | 71–90 | | | | | 1 or 2 | 80% (20/25) | 15.4% (data hidden) [b] | 5,19 | < 0.001 |
| SH2_2019 | | | None | Averagely | | 81–100 | | | | 1 or 2 | >97% (data hidden) [b] | 25% (158/632) | 3.88–4.00 [c] | [c] |
| SH2_2020 | | | None | | 41–100 | | | | | 1 or 2 | 64.8% (46/71) | 25% (126/503) | 2,59 | < 0.001 |
| SH3_2019 | | | | AWNU | | | | | | | 78.3% (517/660) | 64.8% (327/505) | 1,21 | < 0.001 |
| SH3_2020 | | | TMSDI | AWNU | | | | | | | 64.3% (9/14) | 57.2% (218/381) | 1,12 | 0.599 |
| SH4_2019 | F | | | VSU | 71–100 | | | | | | 89.8% (317/353) | 69.7% (83/119) | 1,29 | < 0.001 |
| SH4_2020 | F | 1999 or earlier | | VSAU | | | | | | 1 or 2 | 88.2% (30/34) | 72.1% (320/444) | 1,22 | 0.040 |
| SH5_2019 | F | | | | 41–100 | | | | | | 79.7% (441/553) | 49.5% (45/91) | 1,61 | < 0.001 |
| SH5_2020 | F | 2000–2001 | None | VSU | | | | | | | 82.9% (87/105) | 50% (12/24) | 1,66 | < 0.001 |
| SH6_2019 | M | 2002 or later | None | | 41–100 | | | | | 1 or 2 | 60% (3/5) | 40.1% (139/347) | 1,50 | 0.367 |
| SH6_2020 | M | | None | | 81–100 | | | | | | 64.7% (88/136) | 42.4% (300/708) | 1,53 | < 0.001 |
| SU1_2019 | F | 2002 or later | None or Other MB | | | | | | | | 60% (54/90) | 31.6% (78/247) | 1,90 | < 0.001 |
| SU1_2020 | | | | | 41–100 | | | | | | 44.7% (2344/5248) | 34.3% (390/1136) | 1,30 | < 0.001 |
| SU2_2019 | | 2000–2001 | | | | | | | | | 45.2% (484/1070 | 35.6% (233/655) | 1,27 | 0.010 |
| SU2_2020 | | 2002 or later | | VSU | | | | | | | 42.9% (109/254) | 31.3% (142/454) | 1,37 | 0.002 |
| SU3_2019 | | | None | | 71–90 | | | | | | 89.8% (727/810) | 81.8% (361/445) | 1,10 | < 0.001 |
| SU3_2020 | | | None | | | | | | | 1 or 2 | 78.8% (688/873) | 74.4% (2886/3877) | 1,06 | 0.007 |
| SU4_2019 | | 2002 or later | None | | 1–70 | 41–100 | | | | | 85.7% (6/7) | 38.9% (14/36) | 2,20 | 0.023 |

*(Continued)*

**Table 3.** (Continued)

| Cluster | Sex | Age, based on Y.O.B. | Migration background | Urbanity degree of postal code in 2015 | Income category | Assets category | Parents on social welfare | Parents with SSI | HP parents | Parents working in PSVE sector | Acceptance rate of group ALL* | Acceptance rate of group NONE ** | Ratio accept. rate ALL/ NONE | p-value of the ratio[a] |
|---|---|---|---|---|---|---|---|---|---|---|---|---|---|---|
| SU4_2020 | | | None | | | 41–100 | | | | | 50.9% (799/ 1571) | 35.6% (48/ 135) | **1,43** | **< 0.001** |
| SU5_2019 | M | 2000 or later | | VSU | 71–90 | | | | | | 77.1% (27/ 35) | 32.4% (34/ 105) | **2,38** | **< 0.001** |
| SU5_2020 | M | 2000 or later | Other MB | | | | | | | | 75.6% (34/ 45) | 26.1% (30/ 115) | **2,90** | **< 0.001** |
| SU6_2019 | | | | | 71–100 | | | | | | 81.3% (1077/ 1324) | 73.9% (136/ 184) | **1,10** | **0.017** |
| SU6_2020 | | | | | | | | | | | NA | NA | NA | |
| SU7_2019 | | 2000 or later | None | | | | None | | | | 80.3% (1357/ 1689) | 41.4% (12/ 29) | **1,94** | **< 0.001** |
| SU7_2020 | F | | None | | | 81–100 | | | | | 80.4% (270/ 336) | 54.1 (170/ 314) | **1,49** | **< 0.001** |
| SU8_2019 | | | None | WNU | | | | | | 1 or 2 | 95.3% (41/ 43) | 68.1% (222/ 326) | **1,40** | **< 0.001** |
| SU8_2020 | | | None | WNU | 1–90 | | | | | | 87.1% (101/ 116) | 66% (103/ 156) | **1,32** | **< 0.001** |
| SU9_2019 | M | 2000 or later | | | | | | None | | | 78.4% (344/ 439) | 41.7% (15/ 36) | **1,88** | **< 0.001** |
| SU9_2020 | M | 2000 or later | None or Other MB | | | 41–100 | | None | | 1 or 2 | 76.7% (33/ 43) | <3% (data hidden) [b] | [c] | [c] |

*Acceptance rate group ALL: calculated as the percentage of applicants who were offered a place, out of the total number of applicants in group ALL (Group ALL: the group of applicants who possessed all demographic characteristics positively associated with odds of admission in this cluster's final model)

** Acceptance rate group NONE: calculated as the percentage of applicants who were offered a place, out of the total number of applicants in group NONE (Group NONE: the group of applicants who possessed none of the demographic characteristics positively associated with odds of admission in this cluster's final model)

[a] based on chi-square test of association with $\alpha < 0.05$

[b] data hidden due to CBS data regulations regarding frequencies <10

[c] cannot be calculated or shown precisely, due to CBS data regulations regarding frequencies <10

M = Male; F = Female; Y.O.B. = Year of birth; SSI = Social services income, excluding social welfare; PSVE = primary, secondary or vocational education; HP = registered healthcare professional

TMSDI = Turkish, Moroccan, Surinamese, Dutch Caribbean or Indonesian migration background; MB = Migration background; VSU = (Very) strongly urban; VSAU = (Very) strongly to averagely urban; AWNU = Averagely to weakly/not urban; WNU = Weakly/not urban; NA = not applicable (no significant variables in multivariable logistic regression model)

Note 1: acceptance rates of applicants who had some, but not all, of the characteristics positively associated with odds of admission, fall between the rates of group ALL and group NONE. Due to the high amount of possible combinations of variables, it was not possible to showcase acceptance rates of each type of combination in this table. Information on the acceptance rates of other groups than ALL and NONE can be retrieved from the figures in S1 File. Note 2: If a variable has 3 categories (e.g. A, B, C) and only category B was shown to have significantly different odds of admission compared to reference group A, additional regression analyses were performed with reversed reference categories, to investigate if B and C also significantly differed from each other. The double analyses with different reference categories provided the information which of the three categories had significantly higher odds of admission.

that applicants who were female, were born in 2001 or earlier, had parents in the income category 71–90, had 1 or 2 healthcare professional parents, had no migration background, and who lived in a postal code area which was averagely to weakly/not urban, had significantly higher odds of admission. Cluster SH1_2020 had 25 applicants in total who met *all* six criteria, and 20 of them received an offer of admission. This means group ALL had an acceptance rate of 80%. On the other hand, group NONE had an acceptance rate of 15.4%. As the number of persons accepted in group NONE was smaller than 10, Statistics Netherlands regulations do not allow the publication of the precise frequencies. However, the ratio of 80/15.4 = 5.19 indicates that applicants in group ALL were offered a place more than 5 times as often as applicants in group NONE.

The ratios in Table 3 show that depending on the cluster, group ALL was admitted between 1.06 and 9.01 times as often as group NONE. The study clusters with the highest number of significant variables in the multivariable models are HBO clusters, resulting in the largest difference in acceptance rates between group ALL and group NONE (see Ratio column in Table 3).

To advance our intersectional understanding of (dis)advantage, Fig 2 gives the example of cluster SU4_2020. Two variables were significantly associated with odds of admission: migration background and parental assets category. Acceptance rates differed vastly between groups: 50.1% for applicants without a migration background, 36.4% for TMSDI applicants, and 43.4% for applicants with an other migration background. When we divide these three groups based on parental assets category, we see additional discrepancies. Acceptance rates differed the most between applicants who had a TMSDI migration background and whose parents belonged to the lower assets category (28.8%) versus applicants who had no migration background and whose parents belonged to the higher assets category (50.9%). Those with a TMSDI migration background and parents in the *higher* assets category, had a lower acceptance rate (37.3%) than the applicants without a migration background or with a different migration background from the *lower* assets category (38.7% and 39.8%). In S1 File, similar figures can be found for all clusters. These illustrate how the variables which were significant in the multivariable model, result in (vastly) different acceptance rates for different (intersectional) groups, which diverge from the average acceptance rates as included in Table 1.

The intersectional acceptance rates in S1 File add value to our understanding of inequality of opportunity in the selection procedures for higher education. They show that different groups of applicants (based on a combination of e.g. sex, migration background, and other variables) have very different chances of success in different programs. For example, in cluster SH5_2020 (Creative Media and Game Technologies; Fashion & Textile Technologies; Industrial Design Engineering; and Art and Economics), female applicants from (very) strongly urban areas without a migration background had an acceptance rate of 80.8% (198/245). In comparison, the same cluster only accepted 56.0% of female applicants from (very) strongly urban areas with a TMSDI migration background (28/50). In cluster SU1_2019 (Medicine), female applicants without a migration background had an acceptance rate of 48.4% (1504/3107), while this was only 32.0% for male applicants with a TMSDI migration background (81/253). The latter percentage is significantly lower than the average acceptance rate of the cluster: 45.4%. Combined with the fact that male TMSDI applicants comprised the smallest group of the applicant pool (with only 253 applicants out of 6571 applicants - 3.9%), their low odds of admission contribute to their underrepresentation in the student population. This is part of a pattern found in many clusters. The result is that the student population of many selective programs does not reflect the diversity of the age cohort, the higher general or the pre-university population (see S3 Table).

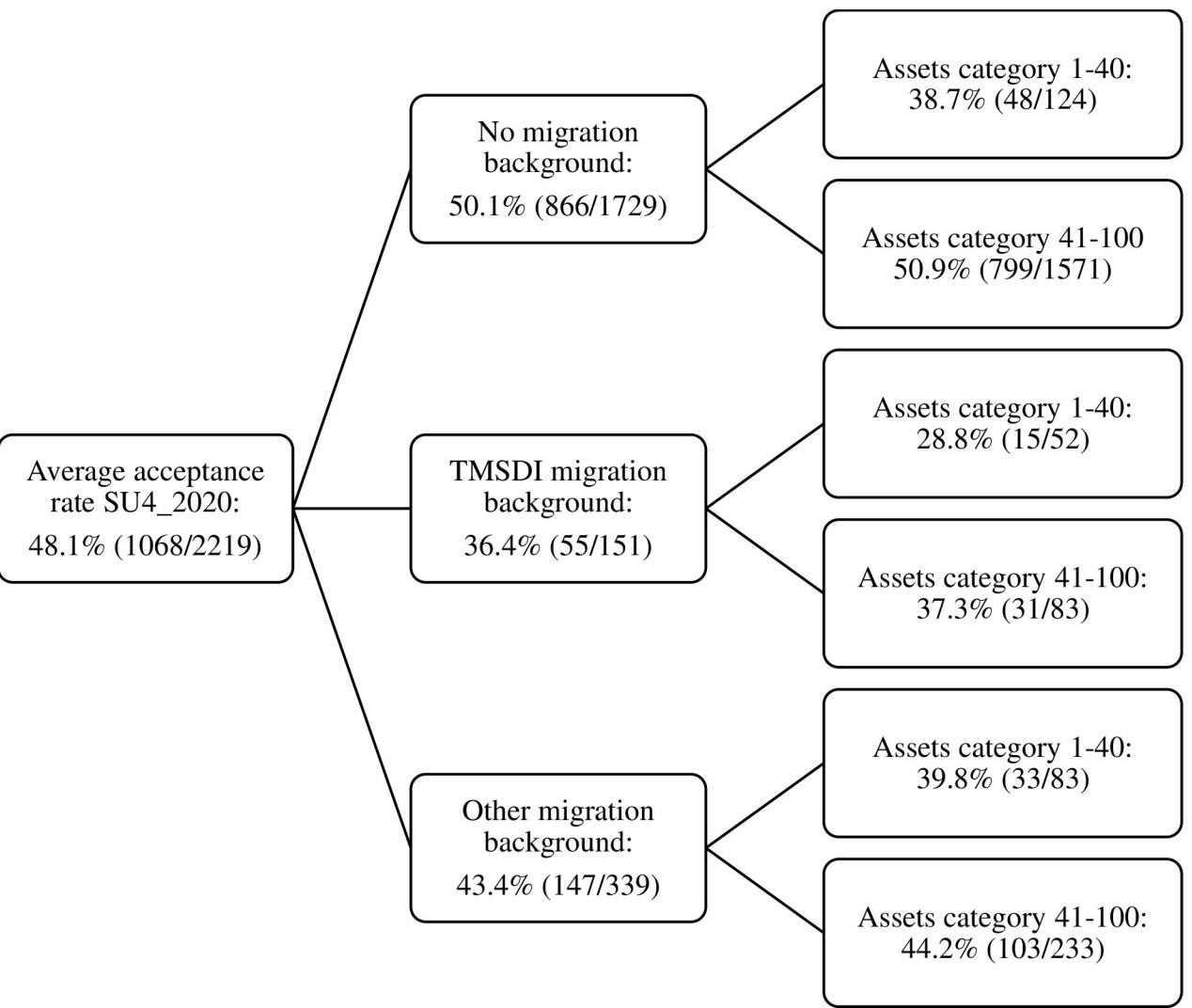

**Fig 2. Intersectional acceptance rates in the cluster SU4_2020 (Biomedical sciences and clinical technology in 2020).** Note: numbers do not always add up due to missing data on parental assets category.

## Discussion

This study aimed to investigate 1) the representativeness of student populations of selective HE programs, compared to the applicant pool and their age cohort; 2) the demographic background variables which are associated with an applicant's odds of admission into selective HE programs; and 3) the acceptance rates of applicants with either *all*, *some* or *none* of the background characteristics positively associated with odds of admission.

The results show that student representativeness in selective HE programs is limited. Part of the reason for this, is self-selection. Applicants are often not representative of their respective eligible pools or the age cohort, especially in the case of sex–as men are underrepresented in the applicant pools of almost every cluster. However, this is not the only reason for the underrepresentation of different groups in selective HE programs. The widespread inequality of opportunity in the selection procedures, both at HBO and university level, contributes to the pattern that student populations do not represent their age cohort, the higher general or

the pre-university population. Only one cluster (Artificial Intelligence and Industrial Design in 2020) did not show significantly different odds of admission based on any of the ten background characteristics of applicants.

The intersectional analyses show that there are vast differences in acceptance rates between group ALL and group NONE (Table 3). Additionally, Fig 2 and the online figures in S1 File show the acceptance rates for groups which possessed some, but not all, of the background characteristics positively associated with odds of admission. Table 1 and the figures prove that the discussion on inequality in selective admissions cannot be based on single background characteristics only: we need an intersectional understanding of inequality. However, the characteristics which resulted in higher odds were not always what one would expect, based on intersectionality theory. For example, men did not always have higher odds of admission, nor did applicants from the highest SES categories or without a migration background. Additionally, whereas international evidence suggests predominantly lower odds for rural applicants [32, 33], this was not always the case in our study: in several study clusters, applicants from weakly to non-urban postal codes had higher odds of admission, whereas in others, their odds were significantly lower. This shows that (dis)advantage is context-dependent. In this section, we discuss the significance and implications of our findings.

The variable which most often related to inequality of opportunity, was migration background. In almost all these cases, applicants with a migration background had significantly lower odds of admission (both at HBO and university programs) than applicants without a migration background. This is problematic, as Dutch and international research shows that children with certain migration backgrounds face numerous obstacles in the educational sphere that limit their potential, starting from a very young age [3, 25, 26, 28]. The inequality based on migration background in admission to HE could not be explained by socio-economic characteristics, as there are several SES indicators included in the multivariable model, which already results in adjusted ORs for migration background. This finding is in line with international research on inequalities based on ethnic background in admission to HE [19, 30, 31].

In clusters where socio-economic indicators influenced the odds of acceptance, applicants from lower and average SES backgrounds were usually (but not always) at a disadvantage. This is disconcerting as well, since these applicants have previously faced several obstacles in and beyond the educational system [55]. As discussed in detail in the results, applicants who do not belong to the highest SES groups are already (strongly) underrepresented in the applicant pools of the majority of clusters.

For the period of 2016–2018, Mulder et al. (2022) found that having healthcare professional parents resulted in significantly higher odds of admission in Medicine, Clinical Technology, Dentistry and Pharmacy. The data presented in this study show that this was no longer the case in 2020 for these programs. However, in other healthcare related study programs, applicants with healthcare professional parents did have significantly higher odds: For HBO programs, these are Allied Medical Care, Nursing, Midwifery, Dental Hygiene, Denturism, Optometry, Physiotherapy, Psychomotoric Therapy/Psychomotricity, and Sport Studies. For university programs, these are Psychobiology and Psychology.

The previously detected inequality based on migration background and sex (Mulder et al., 2022) are no longer found in the 2020 cluster of Medicine (SU1_2020) and Dentistry and Pharmacy (SU2). We speculate that the broad attention towards potential inequality of opportunity in their selection procedures in recent years has led to effective changes to selection procedures. However, it remains important to note that our analyses concern the *nation-wide* applicant pools for these programs. A study by Fikrat-Wevers et al. [56] showed that within individual Medicine, Pharmacy and Clinical Technology faculties, inequality of opportunity in selection still existed. Their multi-site study indicated that men performed significantly poorer

on CVs, but had higher biomedical knowledge test scores than women. On curriculum-sampling tests, applicants with a non-Western migration background had lower scores. CV-scores were lower for first-generation Western immigrants, and significantly lower GPAs were found for first-generation university applicants. Since each program decides the content of their selection procedure, it is important to consider the unique selection context of each HEI. The results of Fikrat-Wevers et al. and this study show that inequality of opportunity in the selection is context-dependent and may change over the years. Therefore, we suggest it is important to replicate our research in future years.

Rejecting low SES applicants and applicants with a migration background results in a loss of potential. Individuals who managed to overcome multiple barriers on the way to reaching the eligibility requirements of HE [26] are definitely talented. Moreover, in some contexts, students from lower SES backgrounds perform better than students who came from higher SES backgrounds and were admitted with the same or a similar GPA [28, 57, 58]. Thus, by rejecting low SES applicants, HEIs lose out on these applicants' contributions to educational quality. Furthermore, society foregoes the opportunity of providing upward socio-economic mobility [59] to members of families who may have had low wealth for generations, which constitutes an economic loss. Society also misses out on the contributions which these applicants could bring to future policymaking and leadership, which are grounded in experiences, knowledge and skills that current leadership from high wealth backgrounds lacks [60].

In our study, the majority of applicants with a migration background are born in The Netherlands (and one or both parents were born abroad). Based on this, these applicants are likely to speak other languages in addition to Dutch and English, which can be useful in the labor market after graduation. For example, most selective programs are related to healthcare (see Table 1). Multilingualism is highly valuable in providing excellent patient care in a diverse society [61]. When a fair share of applicants with a migration background would be admitted, then it is more likely that future healthcare providers are able to speak the languages of the patients they serve. They may also bring with them socio-cultural knowledge that applicants without a migration background may lack. These skills and knowledge bases would be immediately beneficial to culturally sensitive healthcare provision in a multicultural society, and in improving health equity [61, 62]. For example, the study cluster with the highest level of inequality partly consists of Midwifery, for which there are no non-selective alternatives in The Netherlands. This is a healthcare field in which globally, societal inequities and systemic injustice play a large role in unequal outcomes in sexual, reproductive, maternal and newborn health [63]. It is therefore of utmost importance that student populations of these programs become representative of their patient population, in order to contribute to equitable healthcare provision in these areas.

## Recommendations

In '*The Tyranny of Merit*' [14], Michael Sandel discusses why rejecting low SES applicants in favor of applicants from elite backgrounds is an affront to the idea of meritocracy. He therefore proposes the use of lottery-based admission. However, our research shows that compared to the age cohort, applicant pools of many selective programs predominantly consist of students from higher SES backgrounds without an underrepresented migration background (such as Turkish, Moroccan, Surinamese, Dutch Caribbean or Indonesian). This is especially the case for university programs. The lack of a representative applicant pool, especially in the area of migration background and SES, is mainly due to educational inequality on the route to the gates of HE (e.g. lower expectations by the education system [26] in the transition from primary to secondary school), resulting in non-representative higher general and pre-university

student populations. Lottery-based admission will therefore still result in an overrepresentation of high SES students without a migration background, and will not ensure the entry of low SES applicants with an underrepresented migration background [17]. Furthermore, an interview study with Dutch applicants to undergraduate health professions education programs shows that there is no (sub)group of applicants which favors lottery, regardless of whether this is a weighted or a random lottery. They prefer selection instruments on which they can feel more 'in control' [64]. The data in this study suggests that in the Dutch context, *equitable* admissions procedures (which give more opportunities to applicants from underrepresented groups which face significant barriers on the road to HE), have a better chance of increasing the (intersectional) diversity of student populations in selective HE programs than a lottery can achieve. These findings can be relevant in other higher education contexts where applicant pools are also not representative of the general society.

We recommend improved national monitoring in the future of the (potential) inequality of opportunity in HE selection procedures, and its effects on the (intersectional) representativeness of the student population. Furthermore, as Steenman argues, "using selection is generally seen as limiting the standard idea of accessibility and should therefore be used sparingly" [65]. HEIs could consider whether, in the absence of government-mandated capacity limitations, it is justified to use a selection procedure as long as there is no guarantee that this does not harm equality of opportunity and student diversity.

Furthermore, we recommend a detailed study of the selection procedures in the only cluster which did not show any inequality of opportunity, as other selective programs may be able to learn from the Artificial Intelligence and Industrial Design selection procedures in the year 2020. In the same way, study programs in which certain groups are underrepresented (e.g. on the basis of sex or migration background) and/or had lower odds of admission, may be able to learn from other clusters where these underrepresented groups had an advantage in the selection procedure.

**On methodology.** Our findings show that it is important to study inequality of opportunity through an intersectional lens [8]. The acceptance rates of groups ALL versus NONE indicate the vast differences of odds of admission between applicants who either have all the beneficial 'check marks' [60] in a selection procedure, and those who have none. We therefore recommend future quantitative (international) research to build on our methodology, whenever suitable, to create intersectional analyses based on the multiplicity of identity layers that people have. We believe our method to be easier to interpret than traditional regression models with interaction terms. This is especially relevant when the research focuses on inequalities between groups. In order to address and resolve inequities, such as in the educational sphere, it is important that these inequities are clearly demonstrated in a way that all audiences can understand. This is what Table 3 in this article has intended to do.

## Limitations

A limitation is that we could not use parental education levels, as these data are largely missing for parents who were educated outside of the Netherlands. For parents who were educated in The Netherlands, educational achievement is not sufficiently standardized. We circumvented this limitation by including other SES variables that served as proxies.

In the original data, assets percentiles are registered at the household level, on the person who is the main breadwinner. When parents are divorced and live in separate households with partners who each are the main breadwinners, this results in missing data on parental assets.

For a few clusters, parental income and assets data was missing for more than 20% of applicants (namely in SU3, SU7, SU8 and SU9). It is possible that in these cases, applicants were e.g.

internationals who were already living in The Netherlands (giving them a Citizen Service Number), but for whom parental data are missing as parents live abroad. Six out of eight of the multivariable models of these clusters showed no significant association between income/ assets category and the odds of admission. Due to the high amount of missing data, these results must be interpreted with caution.

In this research, we assumed that data was missing completely at random. However, there are methods for handling missing data, e.g. multiple data imputation [66]. In future work, we could investigate the usefulness of multiple data imputation.

Our focus in this paper was to create *association* models for the two academic years under study in the Dutch HE context. These models are not *prediction* models, which could be considered a limitation. Furthermore, the models are not validated in other contexts or other years. To generalize our results to other contexts, we would need to create prediction models which are validated based on data from an external context and from different time periods, which we do not have. Because of the explorative nature of our study, in which our goal was to find out which variables are more or less related to gaining admission to selective programs, we did not adjust the p-values for multiple testing.

Finally, in Table 3, Group ALL and NONE consist in some study clusters of small numbers of applicants and/or admitted students. We addressed this limitation by creating figures similar to Fig 2 for all clusters (in S1 File).

## Future research

The diversity of the student populations as shown in S3 Table only applies to the students at selective programs. As shown in S1 Table, there are many HEIs that do not use selection for the same study programs. Therefore, S3 Table lacks the composition of the student populations of comparable non-selective programs (e.g. in Nursing, where only one out of 17 HBOs used selection). It is possible that self-selection has an effect both on *whether* and *where* students apply to such programs, and that certain groups would rather avoid a selection procedure compared to other groups. Thus, further research could investigate potential differences in student diversity between selective and non-selective programs. This could also aid in mapping the diversity and representativeness of the future workforce in these fields.

Although our quantitative analyses found widespread inequality in selection to Dutch HE, our data cannot explain *what* causes this inequality (e.g. do certain selection instruments disadvantage certain groups? Are certain instruments used more often in some clusters than in others?) and *why* there are such large differences between clusters. Future work could answer these questions with qualitative or mixed-methods research.

Lastly, as we were not able to acquire data on parental jobs for this study other than jobs in the healthcare professions or in education, future research could investigate if other types of parental jobs also have an influence on the odds of admission.

## Supporting information

**S1 Table. Clusters of study programs.**
(DOCX)

**S2 Table. Results from the univariable logistic regression.** *p < .05; **p<0.01; ***p<0.001. Ref. = reference category; PSVA = Primary, Secondary or Vocational or Agrarian; TMSDI: Turkish, Moroccan, Surinamese, Dutch Caribbean, or Indonesian; Urbanity Degree of postal code of residential address in 2015. Results in bold are statistically significant (p < .05). (DOCX)

**S3 Table. Demographic data of all applicants who received a ranking number; all applicants who were offered a seat in the program; and all those who became registered students, for each cluster.**
(XLSX)

**S4 Table. Results of multivariable logistic regression for the final model for each group, performed on applicants eligible for placement.** *p < .05; **p<0.01; ***p<0.001. Ref. = reference category; PSVA = Primary, Secondary or Vocational or Agrarian; TMSDI: Turkish, Moroccan, Surinamese, Dutch Caribbean, or Indonesian; Urbanity Degree of postal code of residential address in 2015. Results in bold are statistically significant (p < .05).
(DOCX)

**S1 File. Intersectional acceptance rates based on background characteristics.**
(DOCX)

**S2 File. Research protocol, including all procedures, sources of variables, and software syntax for statistical analysis.**
(XLSX)

## Acknowledgments

The authors wish to thank all involved people working at Statistics Netherlands, Studielink, the Universities of The Netherlands (UNL) and Vereniging Hogescholen for their cooperation in making this research project possible, and for permitting access to the required datasets. The authors also wish to thank their colleagues at Amsterdam UMC, location VUmc for their valuable feedback on our manuscript. We also thank Ine Vos for her valuable insights on selection procedures and eligibility requirements.

## Author Contributions

**Conceptualization:** Lianne Mulder.

**Data curation:** Lianne Mulder.

**Formal analysis:** Lianne Mulder.

**Funding acquisition:** Lianne Mulder, Gerda Croiset, Rashmi A. Kusurkar, Anouk Wouters.

**Investigation:** Lianne Mulder.

**Methodology:** Lianne Mulder, Eddymurphy U. Akwiwu, Jos W. R. Twisk, Andries S. Koster, Anouk Wouters.

**Project administration:** Lianne Mulder, Anouk Wouters.

**Supervision:** Eddymurphy U. Akwiwu, Jos W. R. Twisk, Andries S. Koster, Jan Hindrik Ravesloot, Gerda Croiset, Rashmi A. Kusurkar, Anouk Wouters.

**Visualization:** Lianne Mulder, Anouk Wouters.

**Writing – original draft:** Lianne Mulder.

**Writing – review & editing:** Lianne Mulder, Eddymurphy U. Akwiwu, Jos W. R. Twisk, Andries S. Koster, Jan Hindrik Ravesloot, Gerda Croiset, Rashmi A. Kusurkar, Anouk Wouters.

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
