## [Decision Letter · Decision Letter 0]

4 Sep 2023

PONE-D-23-06576Inequality of opportunity in selection procedures limits diversity in higher education: An intersectional study of Dutch selective higher education programsPLOS ONE

Dear Dr. Mulder,

Thank you for submitting your manuscript to PLOS ONE. After careful consideration, we feel that it has merit but does not fully meet PLOS ONE’s publication criteria as it currently stands. Therefore, we invite you to submit a revised version of the manuscript that addresses the points raised during the review process.

We look forward to receiving your revised manuscript.

Kind regards,

Stefan Grosek, Ph.D., M.D.,

Academic Editor

PLOS ONE

Journal Requirements:

"This work was supported by Nationaal Regieorgaan Onderwijsonderzoek (NRO), grant number 40.5.18650.007 (https://www.nro.nl/) (awarded to GC as consortium leader), and by a Microdata Access Discount award from ODISSEI, the Open Data Infrastructure for Social Science and Economic Innovations (https://ror.org/03m8v6t10) (awarded to LM)."

5. Please include a caption for figure 1. 

Additional Editor Comments:

Dear Authors

Thank you for submitting your article to the PLOS ONE. It took a while to find appropriate reviewers. Finally we got two very experienced and knowledgable reviewers on the matter you submitted the article. The first author wants to clarify some statistical issues, so please read it carefully and reply accordingly and send it back ASAP.

Kind regards

Reviewers' comments:

Reviewer's Responses to Questions

**Comments to the Author**

1. Is the manuscript technically sound, and do the data support the conclusions?

Reviewer #1: Yes

Reviewer #2: Yes

2. Has the statistical analysis been performed appropriately and rigorously? 

Reviewer #1: Yes

Reviewer #2: Yes

3. Have the authors made all data underlying the findings in their manuscript fully available?

Reviewer #1: Yes

Reviewer #2: Yes

4. Is the manuscript presented in an intelligible fashion and written in standard English?

Reviewer #1: Yes

Reviewer #2: Yes

5. Review Comments to the Author

Reviewer #1: The authors aim to examine how the selection process for higher education programs affects student diversity and representativeness. Specifically, they address: (i) the difference between students in higher education programs compared to the applicant pool, the higher general population, the pre-university population, and their age cohort; (ii) what demographic variables are associated with an applicant's non-admission to selective higher education programs; and (iii) what are the admission rates of applicants with all, some, or none of the background characteristics that are positively associated with the odds of admission.

To answer the research questions, the authors analysed national data from the Netherlands on more than 85,000 applicants for 2019 and 2020.

The first research question was answered by comparing the distributions of the variables of interest between different groups. They highlighted differences between applicants from different clusters of study programs in terms of gender and socioeconomic status.

The second research question was answered using univariable and multivariable logistic regression, separately for different clusters of higher education programs. The authors identified several demographic variables that affect the odds of being accepted into a particular (cluster of) higher education programs and interpreted the differences between clusters of educational programs. Since the authors test a very large number of hypotheses, I wonder if they considered the multiple testing problem and if they evaluated the assumptions of logistic regression. Also, it would be beneficial to include some sort of quasi-R2 to compare the models between clusters of higher education programs.

The third research question was addressed by conducting the "intersectional analysis." For each cluster of study programs, the authors calculated the share of students with matching characteristics with (all/some/none) statistically significant impacts who received an offer of admission. Such a share is called the acceptance rate. Here I do not see much added value of this analysis, and the interpretation of the corresponding results is also limited. For a given example on page 23, it is not clear which groups were considered (for Figure 2; which should perhaps be called Figure 3), and I also think that some figures are not available "insert link after acceptance".

Minor comments:

- The heading levels of, e.g., Migration background and Variables associated with odds of admission should be different.

- The »Variables associated with odds of admission« section should be more clearly separated.

- Could the effect of the covid pandemic be stronger of the more vulnerable groups, and consequently on the selection for HE?

- Missing bracket on page 21, after »final model«.

- Fig. 2 should be Fig. 3 on page 22?

- The term »gender« is used in S3 and »sex« elsewhere.

- Did the authors had in mind »missing completely at random« on page 28?

Reviewer #2: This is a well written contribution on higher education accessibility and unequal opportunities. Data is clearly presented in the text with supporting documents and the text of the contribution is well written and clearly presented. It should be published as it is.

6. PLOS authors have the option to publish the peer review history of their article (what does this mean?). If published, this will include your full peer review and any attached files.

Reviewer #1: No

Reviewer #2: No

---

## [Author Response · Author response to Decision Letter 0]

18 Sep 2023

Our Response to Reviewers document can be found in the files attached to this resubmission. The Table format (with two columns) enables easier reading of our point-by-point response. 

We have copied the contents of the table here as well: 

Reviewer #1

1. Please include a caption for figure 1. 

The caption for Figure 1 can be found on page 10: 

Fig 1. Study size and exclusion criteria

2. The second research question was answered using univariable and multivariable logistic regression, separately for different clusters of higher education programs. The authors identified several demographic variables that affect the odds of being accepted into a particular (cluster of) higher education programs and interpreted the differences between clusters of educational programs. Since the authors test a very large number of hypotheses, I wonder if they considered the multiple testing problem and if they evaluated the assumptions of logistic regression. 

We thank the reviewer for the attention given to our manuscript and their detailed feedback. 

With regard to this question, we would like to respond that our paper is not really about testing hypotheses, it is more dealing with explorative analyses to see which variables are more and which variables are less related to the admission to selective higher education programs. Because of the explorative analyses, we did not adjust the p-values for multiple testing. We added this explanation to the discussion of the paper: “Because of the explorative nature of our study, in which our goal was to find out which variables are more or less related to gaining admission to selective programs, we did not adjust the p-values for multiple testing.”

Regarding the assumptions for (multiple) logistic regression analyses, we evaluated collinearity between the independent variables. Furthermore, all independent variables were either dichotomous or categorical, so the assumption whether the relationship with the independent variable is linear or not, is not applicable.

3. Also, it would be beneficial to include some sort of quasi-R2 to compare the models between clusters of higher education programs. 

Because we were interested in the variables which are related to the admission to selective higher education programs, we did not provide R2 values or any other fit measure for the different multivariable models. As you mentioned, model fit measures provide information about the question which multivariable model is the best. However, which model is the best, does not provide an answer to our research question. We have therefore not included R2 values or any other fit measure for the different multivariable models. 

4. The third research question was addressed by conducting the "intersectional analysis." For each cluster of study programs, the authors calculated the share of students with matching characteristics with (all/some/none) statistically significant impacts who received an offer of admission. Such a share is called the acceptance rate. Here I do not see much added value of this analysis, and the interpretation of the corresponding results is also limited. 

We thank the reviewer for giving us the opportunity to explain the relevance of the intersectional analyses in more detail. The added value of these analyses, is that they make the inequality in opportunity easier to understand for everyone who does not know how to interpret a multivariable logistic regression model. The differences between ‘Group All’ and ‘Group None’ show the level of inequality based on two percentages. In Supplemental Information 5, figures can be found for each cluster, and for the total dataset (2019 and 2020 combined, and separately). We have added a more extensive interpretation to our results with reference to S5 with the following text: 

“The intersectional acceptance rates in Supplemental Information 5 add value to our understanding of inequality of opportunity in the selection procedures for higher education. They show that different groups of applicants (based on a combination of e.g. sex, migration background, and other variables) have very different chances of success in different programs. For example, in cluster SH5_2020 (Creative Media and Game Technologies; Fashion & Textile Technologies; Industrial Design Engineering; and Art and Economics), female applicants from (very) strongly urban areas without a migration background had an acceptance rate of 80.8% (198/245). In comparison, the same cluster only accepted 56.0% of female applicants from (very) strongly urban areas with a TMSDI migration background (28/50). 

In cluster SU1_2019 (Medicine), female applicants without a migration background had an acceptance rate of 48.4% (1504/3107), while this was only 32.0% for male applicants with a TMSDI migration background (81/253). The latter percentage is significantly lower than the average acceptance rate of the cluster: 45.4%. Combined with the fact that male TMSDI applicants comprised the smallest group of the applicant pool (with only 253 applicants out of 6571 applicants, 3.9%), their low odds of admission contribute to their underrepresentation in the student population. This is part of a pattern found in many clusters. The result is that the student population of many selective programs does not reflect the diversity of the age cohort, the higher general or the pre-university population (see Supplemental Information 3).”

5. For a given example on page 23, it is not clear which groups were considered (for Figure 2; which should perhaps be called Figure 3) 

We only have two figures in our manuscript, so we are not able to change this. Figure 1 contains the flow chart with our study size and exclusion criteria. Figure 2 provides an example of the intersectional acceptance rates in the cluster SU4_2020 (Biomedical Sciences and Clinical Technology in 2020).

6. and I also think that some figures are not available "insert link after acceptance". 

We have now provided all the intersectional figures for all clusters in the file in Supplemental Information 5. In this file, we show the acceptance rates of different groups of applicants, based on the variables that were significantly associated with receiving an offer of admission in the corresponding multivariable logistic regression models (see Supplemental Information 4). Within individual clusters, groups are divided based on these characteristics until a division is no longer allowed due to at least one category having less than 10 persons. Statistics Netherlands regulations do not allow publication of frequencies <10. Since a number of clusters had a large number of significant variables in their multivariable models, their figures had to be split up into multiple figures due to limited space on the page. In figures where this is the case, we have added an a, b, or c in the description of the figure.

Example: SH1_2019a is an extension of the figure for SH1_2019. Since this file contains 47 figures in total, we decided that it would be easier for the reader to use the file as it is, rather than having to open 47 individual figures. In the Word file, it is easy to scroll through the figures by using the ‘Navigation pane’ on the left. 

7. The heading levels of, e.g., Migration background and Variables associated with odds of admission should be different. The »Variables associated with odds of admission« section should be more clearly separated. 

We thank the reviewer for their attention to detail and have changed the heading levels accordingly. The headings related to the three research questions (‘Representativeness’, ‘Variables associated with odds of admission’ and ‘Intersectional acceptance rates’) are now in heading level 2, and the variables (Sex; Socio-economic status indicators; migration background) are now in heading level 3. 

8. Could the effect of the covid pandemic be stronger of the more vulnerable groups, and consequently on the selection for HE? 

While it is possible that in individual cases or study programs, the pandemic has played a role in selection outcomes, we have seen no evidence of this in the data on a national level. We made a comparison for all clusters between 2019 and 2020. In 7 clusters, there were more variables significantly associated with the outcome in 2019 than in 2020, meaning that during the pandemic (2020) there were fewer variables of influence in the selection. These clusters are SH1, SH2, SH6, SU1, SU4, SU5, and SU6. 

There were 3 clusters with an equal number of variables influencing the outcome. These clusters are SU3, SU7, and SU8. 

Only in 5 clusters, there were more variables associated with the outcome in 2020 than in 2019. These clusters are SH3, SH4, SH5, SU2, and SU9. 

It is also important to note that in The Netherlands, the lockdown started on March 12, 2020. For a number of selective programs, (some of) the selection instruments (e.g. knowledge tests) had already been completed. This means that the pandemic would have had limited influence on the odds of admission for candidates who had partially or fully completed the procedure. We have therefore not included the potential effect of the pandemic in our manuscript. 

9. Missing bracket on page 21, after »final model«. 

We have added the bracket and thank the reviewer for their attention to detail. 

10. Fig. 2 should be Fig. 3 on page 22? 

We only have two figures in our manuscript, so we are not able to change this. Figure 1 contains the flow chart with our study size and exclusion criteria. Figure 2 provides an example of the intersectional acceptance rates in the cluster SU4_2020 (Biomedical Sciences and Clinical Technology in 2020).

11. The term »gender« is used in S3 and »sex« elsewhere.

We have changed the term in S3 to ‘sex’. 

12. Did the authors had in mind »missing completely at random« on page 28? 

We have changed the text to ‘missing completely at random’ 

Reviewer #2: 

This is a well written contribution on higher education accessibility and unequal opportunities. Data is clearly presented in the text with supporting documents and the text of the contribution is well written and clearly presented. It should be published as it is. 

We thank the reviewer for their compliments.

---

## [Decision Letter · Decision Letter 1]

29 Sep 2023

Inequality of opportunity in selection procedures limits diversity in higher education: An intersectional study of Dutch selective higher education programs

PONE-D-23-06576R1

Dear Dr. Mulder,

We’re pleased to inform you that your manuscript has been judged scientifically suitable for publication and will be formally accepted for publication once it meets all outstanding technical requirements.

Kind regards,

Stefan Grosek, Ph.D., M.D.,

Academic Editor

PLOS ONE

Additional Editor Comments (optional):

Dear Authors

Thank you for your patience and response to the reviewers' comments. I found your article very interesting and imprtant because it addresses a "hidden" problem, not only in your country but also in many others, i.e. inequality of opportunity in selective procedures which limit diversity in higher education as you nicely stated already in your title.

I'll send recommendation to the editor to accept your your article.

Kind regards

Academic Editor

Reviewers' comments:

Reviewer's Responses to Questions

**Comments to the Author**

1. If the authors have adequately addressed your comments raised in a previous round of review and you feel that this manuscript is now acceptable for publication, you may indicate that here to bypass the “Comments to the Author” section, enter your conflict of interest statement in the “Confidential to Editor” section, and submit your "Accept" recommendation.

Reviewer #1: All comments have been addressed

Reviewer #2: All comments have been addressed

2. Is the manuscript technically sound, and do the data support the conclusions?

Reviewer #1: Yes

Reviewer #2: Yes

3. Has the statistical analysis been performed appropriately and rigorously? 

Reviewer #1: Yes

Reviewer #2: Yes

4. Have the authors made all data underlying the findings in their manuscript fully available?

Reviewer #1: Yes

Reviewer #2: (No Response)

5. Is the manuscript presented in an intelligible fashion and written in standard English?

Reviewer #1: Yes

Reviewer #2: Yes

6. Review Comments to the Author

Reviewer #1: (No Response)

Reviewer #2: The manuscript is well prepared and all the issues raised by the reviewers adequately addressed. It should be published.

7. PLOS authors have the option to publish the peer review history of their article (what does this mean?). If published, this will include your full peer review and any attached files.

Reviewer #1: No

Reviewer #2: No

---

## [Editor Report · Acceptance letter]

5 Oct 2023

PONE-D-23-06576R1 

Inequality of opportunity in selection procedures limits diversity in higher education: An intersectional study of Dutch selective higher education programs 

Dear Dr. Mulder:

I'm pleased to inform you that your manuscript has been deemed suitable for publication in PLOS ONE. Congratulations! Your manuscript is now with our production department. 

Kind regards, 

on behalf of

Professor Stefan Grosek 

Academic Editor

PLOS ONE